# Physicians' emotion awareness and emotion regulation training during medical education: a systematic scoping review protocol

Anna Lange ,[1,2] Raphaël Bonvin,[3] Sissel Guttormsen Schär,[4] Sofia C Zambrano[1]

¹Institute of Social Preventive Medicine (ISPM), University of Bern, Bern, Switzerland
²Graduate School of Health Sciences, University of Bern, Bern, Switzerland
³Department of Community Health, University of Fribourg, Fribourg, Switzerland
⁴Universität Bern Institut für Medizinische Lehre, Bern, Switzerland

**Correspondence to**
Anna Lange;
anna.lange@unibe.ch

## ABSTRACT

**Introduction** The objective of this systematic scoping review is to identify what approaches have been implemented in medical education programmes to teach medical students the skills to identify and manage emotions that may be elicited in them during physician–patient interactions and in the clinical environment. Emotions of all involved in the clinical encounter are central to the process of clinical care. However, a gap remains addressing and teaching medical students about recognising and dealing with their own emotions.

**Methods and analysis** This scoping review will follow the updated JBI (The Johanna Briggs Institute) methodology guidance for the conduct and reporting of systematic scoping reviews, and the Preferred Reporting Items for Systematic reviews and Meta-Analyses extension for Scoping Reviews. A search strategy was developed and applied to five databases. Terms used included medical education, medical curriculum, medical students, emotion (regulation), psychological well-being and mental health. Additionally, a grey literature and reference list search will be conducted. Two independent reviewers will first screen titles and abstracts followed by a second, full-text screening phase. Publications to be included will contain information and data about teaching approaches such as lectures, and other teaching material on physicians' emotion awareness and emotion regulation training in medical education.

**Ethics and dissemination** This study will review existing literature on emotion awareness and emotion regulation training in medical education, and a systematic scoping review does not require ethical approval. The results of this scoping review will be submitted for publication to relevant peer-reviewed journals and will be used to inform the development and implementation of training programmes and research studies aimed at preparing medical students to identify and manage their own emotions in the clinical environment.

## INTRODUCTION

A connection between clinicians and patients is central to clinical care.[1] Both, clinicians and patients have emotions, show emotions and will interpret each other's emotions.[2 3] Despite their centrality, the clinicians' own emotions have largely remained unaddressed in medical education and practice,[4] and have often been deemed

## STRENGTHS AND LIMITATIONS OF THIS STUDY

⇒ For standardisation and to meet the quality of other systematic scoping reviews, this scoping review will be conducted in accordance with the JBI methodology for scoping reviews and the Preferred Reporting Items for Systematic reviews and Meta-Analyses extension for Scoping Reviews Checklist.

⇒ To make the screening phase more systematic and the results more reliable, two independent reviewers will perform the first (title and abstract) and second (full-text) screening phases.

⇒ The search strategy is expansive to provide best insight into the field of interest.

⇒ To standardise the extraction of results during the data analysis phase, an extraction tool to chart the data has been developed and will be piloted prior to being used.

as unprofessional,[5–7] leaving clinicians with few skills to identify and manage their emotions. Within emotion management skills, a distinction can be made between 'other-directed' and 'self-directed' emotion skills. 'Other-directed' emotion skills are intended to improve the awareness, understanding and/or management of emotions/emotional states in others, while self-directed emotion skills are aimed to improve awareness, understanding and/or management of emotions/emotional states in the self, such as inward reflection or awareness of own emotions.[8] Since some emotion skills can be allocated to both (e.g., making soothing statements), a clearer distinction can be made according to the intended aim and focus of the emotion skill. To date, the majority of emotion awareness and emotion regulation training in medical education has been focused on emotions directed to the other, for example, emphasising the development of empathy or compassion skills towards patients.[8]

Nevertheless, some competency frameworks of reference for medical training include and expect some self-directed emotion awareness

and other related skills. The CanMEDS framework was first developed in 1990 and serves as a base for many derived frameworks worldwide, foremost describing the competencies which physicians have to develop during their education.[9] In the CanMEDS, in their role as professionals, physicians are expected to have an 'applied capacity for self-regulation, including the assessment and monitoring of one's thoughts, behaviours, emotions and attention for optimal performance and well-being', as well as to have a 'mindful and reflective approach to practice' and 'resilience for sustainable practice'.[9] For the training of medical students, Switzerland recently introduced the PROFILES framework, which is based on a combination of competencies adapted from the CanMEDS framework and Entrustable Professional Activities.[10 11] Here, physician-led regulation and maintenance of the physician's own health, recognising excessive stress or recognising personal illness, as well as being aware of own limits, and seeking supervision, are some of the expected skills that medical students should learn to display as professionals.

Despite these expectations from training, there seems to remain a gap in medical education in addressing and teaching physicians and medical students the skills to identify and manage their own emotions triggered by human interactions within the clinical care environment,[12–15] and to recognise the impact they may have on clinical care or on the clinicians' well-being.

Significant high rates of poor mental health including burnout, emotional exhaustion, depression, anxiety and suicidal ideation are well documented in medical students and in practicing physicians.[16–19] However, the number of preventative interventions and programmes remain sparse and scattered in clinical practice and medical education. Some articles point out factors, such as chronically high workloads or lack of work–life balance which can help explain medical students' and physicians' decreased well-being.[20 21] Other studies point to the important role that emotional blunting, numbness or frequent suppression of own emotions, due to the lack of emotion skills, have on the poor mental health of medical students and physicians.[12–14 22–24]

This scoping review will focus on identifying approaches to teaching medical students about the role of their own emotions in patient care along with the methods employed for self-directed emotion regulation training. To classify the different teaching methods, we will employ the structure developed by Mitchell *et al*,[25] which includes lectures, workshops and other emotion skills training initiatives. For the development of future recommendations and interventions, it is necessary to obtain a comprehensive understanding of current teaching approaches to 'self-directed' emotion regulation training being implemented in medical education. For this purpose, a scoping review is justified and appropriate. A preliminary search of MEDLINE, the Cochrane Database of Systematic Reviews and *JBI Evidence Synthesis* was conducted and no current or underway systematic reviews or scoping reviews on the topic of medical students' management of own emotions or self-directed emotion skills training were

identified. One related work identified has a focus on 'other-directed' emotion skills training in medical education,[8] and therefore the need for an equivalent work completing the picture of self-directed teaching approaches remains. This review will provide invaluable understandings that will facilitate the development of future training initiatives.

## Objectives

The objective of this scoping review is to identify what approaches have been implemented in medical education programmes to teach medical students the skills to identify and manage emotions that they may experience during physician–patient interactions and in the clinical environment in general.

## Research questions

1. What specific teaching approaches, including lectures and training programmes, have been used in medical education to address the emotions physicians experience in the clinical environment?
2. With what teaching methods do specific lectures or related teaching formats cover elements related to physicians' 'self-directed' emotion regulation training during medical education?
3. At what stage during medical education do specific lectures or related teaching formats cover elements related to training physicians to recognise and regulate their own emotions?
4. Which specific aspects of physicians' own emotions, and which specific emotions, are addressed in 'self-directed' emotion regulation training initiatives in medical education?

## Ethics and dissemination

This study will review existing literature on emotion awareness and emotion regulation training in medical education, and a systematic scoping review does not require ethical approval. The results of this scoping review will be submitted for publication to relevant peer-reviewed journals and will be used to inform the development and implementation of training programmes and research studies aimed at preparing medical students to identify and regulate their own emotions in the clinical environment.

## METHODS AND ANALYSIS
### Eligibility criteria

The PCC (Participants, Concept, Context) eligibility criteria of the JBI framework has been used.[26]

## Patient and public involvement

There is no direct patient or public engagement planned for this project. This is a systematic scoping review of the existing peer-reviewed literature on the topic of interest. It is possible, however, that some of the literature included for data analyses might have had patient engagement or public involvement to some degree.

## Participants/population

Studies to be included must report on students during medical school training, excluding those who have obtained the professional degree to practice as physicians and who may have progressed to residency, specialty training or who may have already started to practice as junior doctors. Studies reporting on students from other related fields (e.g., nursing, dentistry or physiotherapy, etc.) are to be excluded.

## Concept

Literature to be included reports on teaching elements of medical programmes, workshops implemented or related implementation formats of content aimed at addressing emotions physicians may experience during clinical encounters or in the clinical environment, as well as 'self-directed' emotion regulation training. Literature is to be excluded if only descriptive of mental health of medical students, only addressing needed changes on the topic of training of physicians' emotions without implementing teaching elements or only reporting on related concepts without addressing elements of physicians' self-directed emotion awareness and emotion regulation skills (e.g., other-directed emotion teaching, relaxation training, etc.).

## Context

To identify teaching approaches, studies to be included must report on the medical teaching environment, either about online or campus-based programmes, seminars, interventions or related teaching formats. International studies from all countries and languages are to be included, no limit back to a certain year will be set to include the literature.

## Types of evidence sources

In this review, we will include experimental and quasi-experimental study designs including randomised controlled trials, non-randomised controlled trials and before and after studies. Exploratory study designs, such as qualitative or mixed-method studies or related study designs will also be included. In addition, studies of any design describing the interventions but not the specific outcomes will also be included. We will exclude books, editorials, comments, posters, and opinion pieces.

## Support

Two research librarians/medical information specialists of the department of the ISPM of the University of Bern supported the development of the search strategy with their expertise. Further consultations with the research librarians are planned throughout the review process.

## Registration

This scoping review protocol is registered in Figshare (10.6084/m9.figshare.23642208). Any adjustments or modifications made throughout the analysis process will be documented and stated in the final scoping review publication. An updated version of the scoping protocol will be available at Figshare.

## Search strategy

This scoping review will be conducted in accordance with the JBI methodology for scoping reviews[26] and the Preferred Reporting Items for Systematic reviews and Meta-Analyses extension for Scoping Reviews (PRISMA-ScR) Checklist.[27] This scoping review protocol follows PRISMA-P guidelines (Moher et al[28]) by employing the items recommended for scoping review protocols by Peters et al.[26] The protocol was developed using the template from the JBI Scoping Review Methodology Group.[29]

After some initial meetings to define the focus of the research strategy and the keywords, the key concepts were chosen, and the final search strategy was peer reviewed with the librarians, using PRESS (Peer Review of Electronic Search Strategies: 2015 Guideline)[30], before it was run. The final key concepts and the search strategy for one database can be found in box 1. For the literature search, a sensitive approach was chosen.[31] The following databases were searched: Medline (Ovid), PsychInfo (Ovid), ERIC (Ovid), CINAHL (EBSCOhost) and Web of Science – Social Sciences Citation Index SSCI (Clarivate, from inception to 15 May 2023). No Language restrictions were applied. A total of 6726 articles were identified. The search strategy combined terms related to the population (medical students, trainees or residents), exposure (medical education, teaching methods (undergraduate), curriculum) and outcomes (e.g., emotion regulation, psychological well-being, mental health). Despite our interest in medical students before obtaining the professional degree to practice as physicians, we included the terms 'trainees' and 'residents' in the search strategy to ensure that studies reporting on mixed populations were not excluded from the outset. In databases where a thesaurus was available, articles were searched by thesaurus terms, in other databases by free text-terms in title and abstract. Duplicate records were removed by using Deduklick,[32] a fully automated deduplication algorithm. The full search strategies of all databases will be included in the appendix of the final scoping review publication.

Additionally, a grey literature search will be conducted, screening relevant literature references found through included full-texts and through hand searches. This search will include books and other sources such as MedEdPortal.org and similar resources. Experts in the field will also be asked for additional publications. An adapted PRISMA 2020 flow diagram describing the individual data sources and its output will be included in the scoping review publication.[33]

## Data extraction

For documenting and managing the search results, the Rayyan software will be used,[34] as well as Endnote.[35] In a first step, identified literature will be screened by the first author and by a second reviewer independently at a title and abstract level. This will be followed by a full-text check of the literature that complied with the inclusion criteria during the first screening process. The full-text screening will also be performed by the two reviewers independently. For piloting purposes of the accuracy and

Box 1    Key concepts for one full search strategy at Medline ALL (via Ovid)

**1) Medical education, teaching methods (undergraduates), curriculum**

(education, medical/ or education, medical, undergraduate/ or exp Curriculum/ or exp Educational Technology/ or (((medical or clinical OR hidden OR informal) adj2 (curricul* or training* or educat* or school or teaching or undergraduate*)) or Entrustable Professional Activit* or 'Situations as Starting Point*' or 'problem-based' or 'project-based*' or ((teaching or training) adj3 (method* or type* or form* or strateg* or intervention* or technique* or setting*)) or academic training* or pedagog* or paedagog* or handbook* or "how to teach" or ((new or innovat* or classroom* or peer* or practice-based or bed-side or bedside or community or interactive* or creativ* or experimental* or precision or reflective) adj3 (teach*)) or 'innovat* method*' or 'diagnostic teaching' or ((small or large or discussion* or project*) adj1 group*) or 'meet the expert*' or tutorial* or 'hands on' or 'psychomotor skills training*' or 'case method*' or 'case study method*' or 'feedback session*' or debat* or quiz* or brainstorm* or lecture* or presentation* or 'hot topic*' or webinar* or seminar* or symposi* or 'case-based stud*' or 'panel discussion*' or video* or movie* or film* or cinema* or 'TED talk*' or 'knowledge test*' or 'facilitat* question* answer*' or 'poster presentation*' or simulation* or 'role-play*' or interview* or 'patient simulat*' or 'flipped classroom*' or forum* or game* or gaming or gamification or 'skill demonstration*' or 'skills demonstration*' or 'mobile teaching app*' or 'social media' or 'scenario based' or 'ward-round*' or 'grand round*' or 'practice exercise*' or 'shadow expert*' or 'inquiry-based' or 'programmed instruct*' or 'study assignment teach*' or 'audiovisual instruction*' or 'direct instruction*' or 'individualised instruction*' or 'individualised instruction*' or microteaching or 'micro-teaching' or 'multimedia instruction*' or scaffolding or 'web based instruction*' or tutoring or workshop* or 'virtual classroom*' or 'group instruction*' or 'formative assessment*' or debrief* or 'information and communication technolog*').ab,ti,kf)

**2) Medical students**

(Students, Medical/ OR (((medical or healthcare or health-care or medicine) adj1 (student* or trainee* or resident*))).ab,ti)

**3) Emotion (regulation), psychological wellbeing, mental health**

(Emotions/ OR Mental Health/ OR (emotion or emotions or feelings or 'emotional intelligen*' or emotionalit* or 'emotional competenc*' or self-aware* or self-regulat* or (emotion* adj3 regulat*) or ((emotion* or intens*) adj2 reaction*) or 'intens* feeling*' or self-reflect* or 'reflective practice*' or self-care* or mindfulness or 'self compassion*' or 'coping strateg*' or psychological wellbeing or psychological well-being or mental health).ab,ti)

**4) Limits (no conference abstracts, letters, etc)**

NOT (letter or news or comment or editorial or congress).pt.

**Translation of search strategy into Medline ALL (via Ovid)**

Segment PPEZ (Ovid MEDLINE Epub Ahead of Print, In-Process & Other Non-Indexed Citations, Ovid MEDLINE Daily, Ovid MEDLINE and Versions 1946 to Present with Daily & Weekly Update)

(education, medical/ or education, medical, undergraduate/ or exp Curriculum/ or exp Educational Technology/ or (((medical or clinical OR hidden OR informal) adj2 (curricul* or training* or educat* or school or teaching or undergraduate*)) or Entrustable Professional Activit* or 'Situations as Starting Point*' or 'problem-based' or 'project-based*' or ((teaching or training) adj3 (method* or type* or form* or strateg* or

*Continued*

intervention* or technique* or setting*)) or academic training* or pedagog* or paedagog* or handbook* or 'how to teach' or ((new or innovat* or classroom* or peer* or practice-based or bed-side or bedside or community or interactive* or creativ* or experimental* or precision or reflective) adj3 (teach*)) or 'innovat* method*' or 'diagnostic teaching' or ((small or large or discussion* or project*) adj1 group*) or 'meet the expert*' or tutorial* or 'hands on' or 'psychomotor skills training*' or 'case method*' or 'case study method*' or 'feedback session*' or debat* or quiz* or brainstorm* or lecture* or presentation* or 'hot topic*' or webinar* or seminar* or symposi* or 'case-based stud*' or 'panel discussion*' or video* or movie* or film* or cinema* or 'TED talk*' or 'knowledge test*' or 'facilitat* question* answer*' or 'poster presentation*' or simulation* or 'role-play*' or interview* or 'patient simulat*' or 'flipped classroom*' or forum* or game* or gaming or gamification or 'skill demonstration*' or 'skills demonstration*' or 'mobile teaching app*' or 'social media' or 'scenario based' or 'ward-round*' or 'grand round*' or 'practice exercise*' or 'shadow expert*' or 'inquiry-based' or 'programmed instruct*' or 'study assignment teach*' or 'audiovisual instruction*' or 'direct instruction*' or 'individualized instruction*' or 'individualised instruction*' or microteaching or 'micro-teaching' or 'multimedia instruction*' or scaffolding or 'web based instruction*' or tutoring or workshop* or 'virtual classroom*' or 'group instruction*' or 'formative assessment*' or debrief* or 'information and communication technolog*').ab,ti,kf) AND (Students, Medical/ or (((medical or healthcare or health-care or medicine) adj1 (student* or trainee* or resident*))).ab,ti) AND (Emotions/ or Mental Health/ or (emotion or emotions or feelings or 'emotional intelligen*' or emotionalit* or 'emotional competenc*' or self-aware* or self-regulat* or (emotion* adj3 regulat*) or ((emotion* or intens*) adj2 reaction*) or 'intens* feeling*' or self-reflect* or 'reflective practice*' or self-care* or mindfulness or 'self compassion*' or 'coping strateg*' or psychological wellbeing or psychological well-being or mental health).ab,ti) NOT (letter or news or comment or editorial or congress).pt

sensitivity of the inclusion and exclusion criteria, 20% of the papers will be screened by the first and last author of this protocol. In case of disagreement at any of the screening phases, discussions will be held between the two screeners, and when needed, a third reviewer will be involved to reach an agreement.

Of the full-texts selected in the second screening phase, data will be extracted for analysis. To chart the data, an extraction tool (table 1) has been developed and will be piloted to test its feasibility. After the piloting, possible adaptations of the extraction tool will be made, and the final completed extraction tool will be published in the scoping review. As the studies to be included may contain interventions, the extraction tool includes a combination of applicable items from the TIDieR (Template for Intervention Description and Replication) and the PAGER (Patterns, Advances, Gaps, Evidence for practice, Research recommendations) frameworks, as well as additional variables added by the authors. Data will be extracted using Covidence, a web-based software for systematic reviews, to allow for collaboration, tracking of changes and secure storage. Once data extraction has been completed, the datasheet will be imported for analyses into MAXQDA.[36]

**Table 1** Extraction tool

| Item category | Item |
|---|---|
| Study characteristics / patterns | Study DOI |
| | Title of publication |
| | Year of publication |
| | Name(s) of author(s) |
| | Email address corresponding author |
| | Aim(s) of the research study |
| | Country in which study was conducted |
| | Name of medical school/institution where teaching was held |
| | Year(s) in which study was conducted |
| | Short description of year(s)/period in which study was conducted |
| | Type of study design |
| | Emotions topic |
| | Healthcare area |
| Participants | Participant numbers total included (table) |
| | Age participant numbers total included (table) |
| | Participation numbers (completed intervention) (table) |
| | Age participation numbers (completed intervention) (table) |
| | Method of recruitment of participants (selection of: phone, mail, email, medical clinic, during lecture/course, not reported, other) |
| TIDieR framework and additional items added by the authors/advances | Aim of intervention |
| | Brief name of intervention |
| | Main competencies addressed in the intervention/main competencies taught |
| | Specific context of intervention |
| | Procedures: each of procedures/activities and/or processes used in intervention (including enabling or support activities) |
| | Who provided the intervention |
| | Modes of delivery |
| | Delivery provided individually or in group? (selection of: group, individually, other) |
| | Type(s) of location(s) for intervention (including infrastructure, etc) |
| | Number of interventions |
| | Number of sessions intervention 1 was delivered |
| | Number of sessions intervention 2 was delivered |
| | Number of sessions intervention 3 was delivered |
| | Number of sessions intervention 4 was delivered |
| | Period of time intervention 1 was delivered |
| | Period of time intervention 2 was delivered |
| | Period of time intervention 3 was delivered |
| | Period of time intervention 4 was delivered |
| | Schedule |
| | Duration of session for intervention 1 |
| | Duration of session for intervention 2 |
| | Duration of session for intervention 3 |

Continued

**Table 1** Continued

| Item category | Item |
|---|---|
| | Duration of session for intervention 4 |
| | Who received? (year of study, which students?) (open text field) |
| | Who received the intervention (selection box) (selection of: medical students during non-clinical years, medical students during clinical years, mixed medical students and postgraduates, mixed medical students and undergraduates (undergraduate: not at medical school yet), unclear, other) |
| | Mandatory or elective training? (selection of: mandatory, elective, not reported, other) |
| | Tailoring planned? (selection of: yes, no, other) |
| | If yes: what, why, when, how (for each modification) |
| | Modifications to intervention during course of study? (yes, no, other) |
| | If yes (which modifications during course of study) (what, why, when, how/per for each modification) |
| Outcomes | Outcomes measured (yes, no, other) |
| | Outcomes assessed |
| | (Significant) results/conclusions/outcomes |
| | Was programme evaluated? (yes, no, other) |
| Remaining part of PAGER framework that was not yet covered by the previous items | Gaps/research recommendations |
| | Evidence for practice |
| | Limitations of the study (designs) reported by the author(s)/challenges in training that may explain results |
| Source of funding | Details source of funding |

The rows of this table represent the columns of the extraction tool to be used.

## Data analyses and presentation

The analysis of the data will be narrative and descriptive, presenting the results in the forms of diagrams and tables and describing the literature included in terms of conceptual categories such as intervention type, duration of intervention, timing of the intervention across the curriculum, key findings, research gaps, as well as related categories relevant to the study questions.

## DISCUSSION

This scoping review will provide an overview of the approaches used in medical education to teach physicians on identifying and managing the emotions they may experience during physician–patient interactions or in the clinical environment, including 'self-directed' emotion regulation training. The review could help identify common themes, formats and methods used to teach medical students about how to be aware of and how to manage their own emotions, the effectiveness of these initiatives, and identify gaps in the

curriculum of medical students in that very field of training and education for their future as physicians. Ultimately, the results of this scoping review could provide recommendations to inform the future implementation of training programmes in medical education to address physicians' own emotions in the clinical environment.

**Acknowledgements** The authors thank the librarian Beatrice Minder from the University Library of Bern for her support and help with designing the search strategies and Doris Kopp-Heim from the University Library of Bern for the peer-review of the searches. The authors would also like to thank Lina Murer, master's student in Psychology, for supporting this project and for being the second independent screener of the literature screening process. This scoping review is to contribute towards the PhD degree of the first author (AAL).

**Contributors** AAL and SCZR conceived the idea of the study and developed the initial design of the scoping review. AAL, RB, SG and SCZR contributed to the methodological design of the review. AAL and SCZR drafted the manuscript. All authors critically revised the manuscript for important intellectual content. SCZR is the guarantor of the systematic scoping review.

**Funding** This scoping review is being funded by the the Schweizerischer Nationalfonds zur Förderung der Wissenschaftlichen Forschung (Swiss National Science Foundation (SNSF)) via an Eccellenza Professorial Fellowship grant to the last author (SCZR) (grant number PCEFP1_194177). The funder has had no role in the development of this protocol and will not be involved in any of the scientific steps required to complete the scoping review.

**Competing interests** None declared.

**Patient and public involvement** Patients and/or the public were not involved in the design, or conduct, or reporting or dissemination plans of this research.

**Patient consent for publication** Not applicable.

**Provenance and peer review** Not commissioned; externally peer reviewed.

**ORCID iD**
Anna Lange http://orcid.org/0000-0002-8150-3712

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
