## [Reviewer comments · BMJ Open]

ARTICLE DETAILS

TITLE (PROVISIONAL)	Physicians' emotion awareness and emotion regulation training during medical education: a systematic scoping review protocol
AUTHORS	Lange, Anna; Bonvin, Raphaël; Guttormsen, Sissel; Zambrano Ramos, Sofia

VERSION 1 – REVIEW

REVIEWER	Del Piccolo, Lidia University of Verona, Department of Neurosciences, Biomedicine and Movement Sciences
REVIEW RETURNED	22-Oct-2023

GENERAL COMMENTS	The proposed systematic scoping review is interesting and the described methods to conduct analyses are appropriate. I have just three elucidating questions: P. 7 in the "Concept" paragraph, authors report in the examples (line 46) that papers reporting relaxation training are not to be considered among the physiscans' self-directed emotion skills, why? P. 9 line 23 why are trainees and residents included if in the participants/population they explicitly stated that "Studies to be included must report on students during medical school training, excluding those who have obtained the professional degree to practice as physicians and who may have progressed to residency, specialty training, or who may have already started to practice as junior doctors."? P. 9 line 26 (I would add to emotion regulation also emotion awarness)
---

REVIEWER	Weurlander, Maria Stockholms Universitet, Dept of Education
REVIEW RETURNED	13-Nov-2023

GENERAL COMMENTS	This paper outlines a protocol for a scoping review on teaching approaches related to emotion skills training for medical students. It is an interesting and crucial paper as there is an increasing body of research documenting emotional challenges and distress among medical students. The introduction focuses on pertinent previous studies and clearly outlines the research gap and the contribution the scoping review will make. The objectives and research questions are relevant. However, I am curious as to why the authors do not explicitly mention the outcomes of the teaching approaches they will review in the research questions. Would it not be worthwhile to explore whether and how the described teaching approaches (in the
---

	reviewed papers) are successful in imparting emotional skills to medical students? If this is not included in the review, please clarify the rationale behind this decision for the readers. The authors detail the review process, including inclusion criteria, in the methods section. The search strategy, developed and employed, is presented in detail. The data extraction procedure is also adequately outlined. However, in Table 2, under the PAGER framework, the authors mention that they will seek evidence for practice. This raises a question about what is being referred to in this context. This is particularly relevant to my earlier query about whether the outcomes of the studies included in the review will be examined.
--	---

VERSION 1 – AUTHOR RESPONSE

Name Editor/Reviewer	Comments	Authors' response	Action taken
Prof. Lidia Del Piccolo Comment 3	P. 7 in the "Concept" paragraph, authors report in the examples (line 46) that papers reporting relaxation training are not to be considered among the physicians' self-directed emotion skills, why?	Thank you very much for this remark Prof. Del Piccolo. Studies that are about relaxation alone and which do not make a link to the importance of relaxation for future patient encounters will be excluded, as the focus of this review is the teaching of skills with the rationale of the future need during consultations with patients. It is the same reason why studies reporting on academic stress are also being excluded. However, if an article reported on relaxation training in the context of patient interactions, the article would be included.	We have rewritten the sentence to clarify that without the link to patient encounters, specific types of emotion training will be excluded (now on page 7/8).

Prof. Lidia Del Piccolo Comment 4	P. 9 line 23 why are trainees and residents included if in the participants/population they explicitly stated that "Studies to be included must report on students during medical school training, excluding those who have obtained the professional degree to practice as physicians and who may have progressed to residency, specialty training, or who may have already started to practice as junior doctors."?	Thank you very much for this question. Although this approach yielded more results, we wanted the search to be as comprehensive as possible, since studies sometimes mix the populations and we didn't want to miss studies which the other terms wouldn't have been sensitive enough to identify.	We added a sentence in the manuscript to clarify our approach: see page p. 9.
Prof. Lidia Del Piccolo Comment 5	P. 9 line 26 (I would add to emotion regulation also emotion awareness)	Thank you very much for your comment and input! As it is not an exhaustive list of all the terms we employed in the search string, we have added some more keywords, but also clarified that these are examples. As in the search string can be seen on page 10 (3rd point in the table), we included the terms "...self-aware*", or self-regulat* or (emotion* adj3 regulat*)...".	We included "e.g." on page 9: "(e.g." emotion regulation, psychological wellbeing, or mental health)".
Dr. Maria Weurlander Comment 6	The introduction focuses on pertinent previous studies and clearly outlines the research gap and the contribution the scoping review will make. The objectives and research questions are relevant. However, I am curious as to why the authors do not explicitly mention the outcomes of the teaching approaches they will review in the	Thank you for your feedback and input Dr. Weurlander. Our goal with this scoping review was to include as many teaching approaches as possible and therefore decided to not limit the selection of studies to those with more experimental designs in which outcomes are reported. As it would still be important to know some of the effects of those interventions/teaching approaches which have been tested, we have adapted the extraction sheet. The "effectiveness of these initiatives" was planned to be reported on, but we had not included it in the extraction tool so explicitly.	We have adapted the extraction sheet to include the study outcomes, as well as other important details of the teaching approaches which we had not explicitly listed in the initial draft. Please refer

	research questions. Would it not be worthwhile to explore whether and how the described teaching approaches (in the reviewed papers) are successful in imparting emotional skills to medical students? If this is not included in the review, please clarify the rationale behind this decision for the readers.		to page No 13.
Dr. Maria Weurlander Comment 7	The authors detail the review process, including inclusion criteria, in the methods section. The search strategy, developed and employed, is presented in detail. The data extraction procedure is also adequately outlined. However, in Table 2, under the PAGER framework, the authors mention that they will seek evidence for practice. This raises a question about what is being referred to in this context. This is particularly relevant to my earlier query about whether the outcomes of the studies included in the review will be examined.	Thank you for this comment. "Evidence in practice" is part of the PAGER framework. In describing what is included, the authors of the framework state that even though scoping reviews do not seek to report on the quality of evidence, they argue that many scoping reviews fall short on providing useful messages for practice. They therefore support for scoping reviews to use this item to provide a (broad) interpretation of 'practice' as a way of delivering the practical message that could be extracted from the literature. This can be for example in form of implications for participants, patients, physicians, etc. They state that providing this information is important for demonstrating the utility of the review (and the associated work effort), as opposed to remaining at a descriptive level of themes. This, combined with the addition of the outcomes to be extracted (see previous comment), will ensure that our review is still contributing some evidence for practice. Pager reference: Bradbury-Jones, 2021 https://doi.org/10.1080/13645579.2021.1899596	We have now updated the extraction form to describe what will be extracted from each of those items, as well as to include the extraction of outcomes of those studies for which that item would apply to.

VERSION 2 – REVIEW

REVIEWER	Del Piccolo, Lidia University of Verona, Department of Neurosciences, Biomedicine and Movement Sciences
REVIEW RETURNED	20-Feb-2024
GENERAL COMMENTS	I suggest a slight change in the title to make it more straightforward and not misleading: "Physicians' emotion awareness and emotion regulation training during medical education: a systematic scoping review protocol"

VERSION 2 – AUTHOR RESPONSE

Name Editor/Reviewer	Comments	Authors' response	Action taken
Prof. Lidia Del Piccolo Comment 1	I suggest a slight change in the title to make it more straightforward and not misleading: "Physicians' emotion awareness and emotion regulation training during medical education: a systematic scoping review protocol"	Thank you very much for this comment. We agree that the adjustment of the title will make it clearer for the reader. We are happy to implement your suggestion as our new title.	We adjusted the title (see p. 1) and adopted the terms "emotions awareness and emotion regulation" within the manuscript, to replace the terms "emotions and emotion skills" when needed for its consistency with the title. (see tracked changes throughout the document).